# Optimized active layer morphology toward efficient and polymer batch insensitive organic solar cells

Kangkang Weng[1], Linglong Ye[1,2], Lei Zhu [3], Jinqiu Xu[3], Jiajia Zhou [1], Xiang Feng[4], Guanghao Lu [4], Songting Tan[2], Feng Liu [3✉] & Yanming Sun [1✉]

Morphology control in laboratory and industry setting remains as a major challenge for organic solar cells (OSCs) due to the difference in film-drying kinetics between spin coating and the printing process. A two-step sequential deposition method is developed to control the active layer morphology. A conjugated polymer that self-assembles into a well-defined fibril structure is used as the first layer, and then a non-fullerene acceptor is introduced into the fibril mesh as the second layer to form an optimal morphology. A benefit of the combined fibril network morphology and non-fullerene acceptor properties was that a high efficiency of 16.5% (certified as 16.1%) was achieved. The preformed fibril network layer and the sequentially deposited non-fullerene acceptor form a robust morphology that is insensitive to the polymer batches, solving a notorious issue in OSCs. Such progress demonstrates that the utilization of polymer fibril networks in a sequential deposition process is a promising approach towards the fabrication of high-efficiency OSCs.

[1] School of Chemistry, Beihang University, 100191 Beijing, PR China. [2] Key Laboratory of Environmentally Friendly Chemistry and Applications of Ministry of Education, College of Chemistry, Xiangtan University, Xiangtan 411105, PR China. [3] Frontiers Science Center for Transformative Molecules, School of Chemistry and Chemical Engineering, Shanghai Jiao Tong University, Shanghai 200240, PR China. [4] Frontier Institute of Science and Technology, Xi'an Jiaotong University, Xi'an 710054, PR China. ✉email: fengliu82@sjtu.edu.cn; sunym@buaa.edu.cn

Over the past few decades, bulk-heterojunction (BHJ) organic solar cells (OSCs) have attracted tremendous attention due to their intrinsic advantages of lightweight, low cost, and the fabrication of flexible and large-scale devices via solution processing[1–13]. Achieving an ideal BHJ morphology that can balance exciton dissociation and charge transport is a prerequisite for achieving high-efficiency OSCs[14–21]. During the film deposition, the drying kinetics plays a critical role in the formation of the BHJ morphology, which dictates the final nanostructure of crystalline and mixed region[22–27]. Blend casting (BC) (e.g., spin coating) is routinely used to fabricate BHJ films on a laboratory scale. The thin film is quickly formed and the morphology is locked[28,29]. In terms of printing fabrication, the solvent vaporizes slower compared with spin coating method, which makes morphology more unpredictable due to the different thermodynamics and kinetics of materials in crystallization and phase separation under prolonged time scales[30]. Hence, controlling the BHJ morphology under different film-forming processes (the lab-scale spin coating versus industrial printing) remains challenging for the commercialization of OSCs.

The two-step sequential deposition (SD) method provides an avenue to control the active layer morphology, in which the donor and the acceptor are deposited sequentially[31–35]. The first layer undergoing simple solution evaporation induced crystallization only, which can eliminate complicated materials interactions and phase separation. The crystalline feature can be easily handled by the choice of solvent and drying time. The second layer morphology can be controlled by fundamental properties of crystallization and miscibility with the first layer material. The swelling induced diffusion of second layer materials into the first one is highly manageable by choosing proper solvent and drying speed[35]. Such a thin film fabrication, in couple with the proper first layer construction, can be highly useful in morphology control.

We have recently demonstrated a fibril network strategy (FNS) to control the BHJ morphology of OSCs[36–39]. In FNS, the polymer fibrils offer a high-speed channel for hole transport, and the small molecule acceptors (SMA) reside in-between the fibril network to perform electron transport. The ideal BHJ interpenetrating network structure can be thus fabricated. The coupling of FNS with SD, simply using a well-established polymer fibrillar network in the first layer, finds a route to optimize the morphology framework, in which non-fullerene acceptor molecules can diffuse into the better-developed fibril mesh to form a bicontinuous network morphology. The proof of concept is achieved using conjugated polymer PT2 as the donor that can self-assemble into well-defined fibril structure in neat film and SMA Y6[40] as the acceptor that can be readily introduced into bottom layer via swelling. The two-step process can be easily optimized, and a favorable morphology is achieved, where the crystalline fiber structure of PT2 remains intact and amorphous region mixed well with Y6 acceptor. The prescribed morphology guarantees a large interfacial area between PT2 and Y6 and high efficient transport network. As a result, a high power conversion efficiency (PCE) of 16.5% (certified as 16.1%) is achieved, which is the highest PCE values reported for SD OSCs so far. Such method is implemented in large-area device fabrication using slot die printing, and a PCE of 14.6% is obtained for the device with an active area of 0.8 cm². The FNS-SD fabrication method highlights the full development of material properties and reduction of uncontrollable interactions, which shows high material tolerance. Varied PT2 polymers are synthesized with different molecular weight and polydispersity values, and FNS-SD processing leads to highly uniform device performance, solving the notorious batch-to-batch variation problem in OSCs.

## Results

**Photovoltaic performance.** The chemical structures of PT2 and Y6 are shown in Fig. 1a. As shown in Fig. 1b, for the SD process, PT2 is deposited from chlorobenzene (CB) solution to form a fibrous mesh film. Chloroform (CF) is chosen as the casting solvent for Y6 to swell the PT2 layer and assist Y6 molecules to diffuse into the fibril mesh. 1,8-diiodooctane (DIO) solvent additive and thermal annealing treatment were employed to further optimize the active layer morphology. The optimal content of DIO solvent additive and the thermal annealing temperature were found to be 1% and 100 °C, respectively. A conventional device structure of ITO/PEDOT:PSS/active layer/PNDIT-F3N-Br/Ag was used in this study.

As presented in Supplementary Fig. 4, PT2 displays the main absorption range from 400–700 nm, which is complemented well with Y6. A high photocurrent is expected for OSCs based on PT2:Y6 blend. For the fabrication of SD devices, we first optimized the thicknesses of PT2 and Y6 (Supplementary Table 1). The optimal thicknesses of PT2 and Y6 were found to be 50 nm and 60 nm, respectively. OSCs based on SD processed film without any treatments showed a PCE of 12.4%, with an open-circuit voltage ($V_{oc}$) of 0.87 V, a short-circuit current ($J_{sc}$) of 24.7 mA cm⁻², and a fill factor (FF) of 57.6% (see Fig. 1c and Table 1). DIO solvent additive is introduced into Y6 solution, which vaporizes much slower comparing to solvent and thus help to improve Y6 crystallization and diffusion. Better device performance was thus obtained with a combination of DIO additive and thermal-annealing treatment (Supplementary Table 2). As a result, the corresponding device shows a $V_{oc}$ of 0.83 V, a $J_{sc}$ of 26.7 mA cm⁻², a FF of 74.4%, and a PCE of 16.5%, which is the highest PCE value reported for SD OSCs to date. The PCE was certified by NIM, China, using a 3.152 mm² mask, and a PCE of 16.1% was recorded (Supplementary Fig. 5). BC OSCs based on PT2:Y6 blend are fabricated as control device for comparison. As shown in Table 1 and Supplementary Table 3, the best PCE of 15.0% was obtained, with a $V_{oc}$ of 0.83 V, a $J_{sc}$ of 26.3 mA cm⁻², and a FF of 68.9%. The external quantum efficiency (EQE) spectra of OSCs were presented in Fig. 1d. The high photo response in the wavelength range of 350–850 nm observed for SD OSCs agreed well with the high $J_{sc}$ value from $J$–$V$ measurements. In addition, we monitored the storage stability of the optimum SD OSCs, as shown in Supplementary Fig. 6. After exposed in air for more than 300 h, the encapsulated SD device maintained 98.8% of its initial PCE, which is higher than that of the BC device. The results indicate that the prescribed fibril network BHJ morphology in SD fabrication is superior than that of BC processing, which results from the simple thermodynamics that materials interaction is less intangled with crystallization, and diffusion of Y6 into amorphous PT2 creates better quality n-type regions.

Space charge limited current (SCLC) method was used to investigate the charge transport property in the blend films (Supplementary Fig. 7 and Table 4). The as-cast SD device shows relatively low hole/electron mobilities of $(1.3 ± 0.1) × 10^{-4}/(2.3 ± 0.2) × 10^{-4}$ cm² V⁻¹ s⁻¹. After using DIO additive and thermal annealing treatments, both hole and electron mobilities of the SD device were enhanced, and a balanced hole/electron mobilities of $(5.9 ± 0.2) × 10^{-4}/(5.4 ± 0.2) × 10^{-4}$ cm² V⁻¹ s⁻¹ can be achieved. In comparison, the optimal BC device showed a hole mobility of $(2.5 ± 0.2) × 10^{-4}$ cm² V⁻¹ s⁻¹ and an electron mobility of $(3.5 ± 0.3) × 10^{-4}$ cm² V⁻¹ s⁻¹. In addition, photo-induced charge carrier extraction by linearly increasing the voltage (photo-CELIV) was also employed. Supplementary Fig. 8 shows the photo-CELIV curves of SD and BC devices. The carrier mobility of the SD device is $1.02 × 10^{-4}$ cm² V⁻¹ s⁻¹, which is much higher than that $(4.1 × 10^{-5}$ cm² V⁻¹ s⁻¹) of the BC OSC.

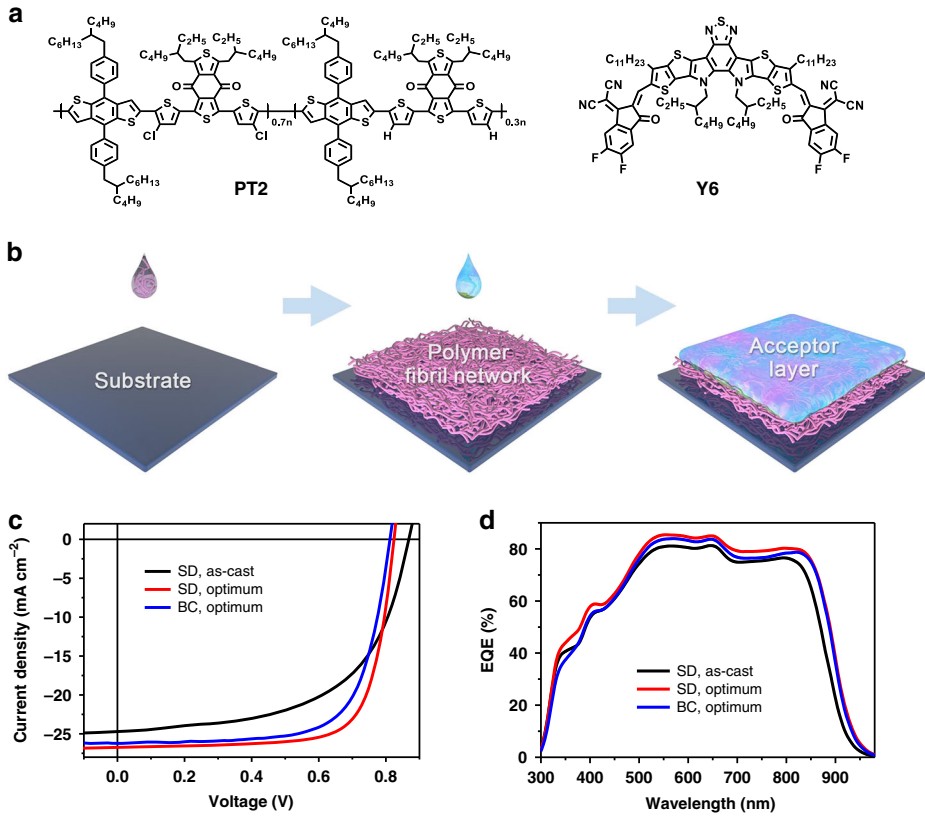

**Fig. 1 Chemical structures, sequential deposition process, photovoltaic performance. a** Chemical structures of PT2 and Y6. **b** Schematic illustration of fabrication procedure of SD blend film. **c** J–V characteristics of SD and BC OSCs under constant incident light intensity (AM 1.5 G, 100 mW cm$^{-2}$), and **d** the corresponding EQE spectra.

### Table 1 Summary of device parameters of SD and BC OSCs.

| Device | $V_{oc}$ (V) | $J_{sc}$ (mA cm$^{-2}$) | FF (%) | PCE (%) (average)[a] |
|---|---|---|---|---|
| SD (as-cast) | 0.87 | 24.7 (24.2)[b] | 57.6 | 12.4 (12.3 ± 0.1) |
| SD (optimum) | 0.83 | 26.7 (26.2) | 74.4 | 16.5 (16.3 ± 0.2) |
| BC (optimum) | 0.83 | 26.3 (25.6) | 68.9 | 15.0 (14.8 ± 0.2) |
| SD (optimum) | 0.81 | 26.3 | 75.1 | 16.1[c] |

[a]The average parameters were calculated from 10 independent cells.
[b]EQE values.
[c]Certified by National Institute of Metrology, China (NIM).

The high and balanced charge transport partially accounts for the high $J_{sc}$ and FF achieved in the optimized SD OSCs.

Charge generation, and charge extraction were studied by analyzing the photocurrent density ($J_{ph}$) *versus* the effective voltage ($V_{eff}$)[41]. $J_{ph}$ is defined as $J_{ph} = J_L - J_D$, wherein $J_L$ and $J_D$ represent the current densities under illumination and in the dark, respectively. $V_{eff}$ is defined as $V_0 - V_a$, wherein $V_0$ is the voltage when $J_{ph}$ is 0. $V_a$ represents the applied voltage bias. The ratio of $J_{ph}/J_{sat}$ can present the efficiency of exciton dissociation and charge collection, wherein $J_{sat}$ is the saturation photocurrent density. The $J_{ph}/J_{sat}$ ratio for the optimized SD device was calculated to be 95.0%, (Supplementary Fig. 9), under the maximum output power condition. In contrast, the BC device showed a $J_{ph}/J_{sat}$ ratio of 91.3%.

**Morphology investigation.** Vertical phase separation is recorded in film-depth-dependent light absorption spectra, which is a facile approach to investigate the variations of composition, crystalline ordering, electronic/optical properties in the vertical direction[42,43]. According to the neat film absorption spectra of the two materials, we focused on the characteristic absorption peak with PT2 of 626 nm, and Y6 of 804 nm. We first fabricated a layer-by-layer film by transfer printing. As shown in Supplementary Fig. 10, such films inevitably show vertical phase distribution due to the limited material diffusion. The current choice of sequential deposition with the right choice of solvent that distinguishes the solubility nonequilibrium toward amorphous and crystalline materials brings in handle in morphology manipulation. The film-depth-dependent light absorption spectra of the as-cast SD, optimum SD and BC blend films were illustrated in Fig. 2a–c. Different from the results from previous reports that SD films have a vertical phase distribution with a large amount of acceptors on top surface and a large amount of donors on bottom surface thereof[44]. Homogeneous distributions of PT2 and Y6 in the vertical phase are found in both as-cast and optimum SD thin films. Here, we used chloroform as the casting solvent to deposit the top layer, which could quickly introduce Y6 into PT2 mesh network. Thus a uniform morphology is formed. Adding a small amount of DIO additive that vaporizes much slower than chloroform could increase the degree of crystallinity of PT2 and improves the morphology and transport, to achieve high performance.

Film-depth-dependent light absorption were further used to investigate the crystalline ordering of PT2 at different film-depth position. As shown in Fig. 2a–c, the absorption peak of PT2 is independent on film depth, implying that its HOMO level (also hole transport level) is invariable along charge transport direction (film-depth direction). This negligible variation of transport level is beneficial to avoid trapping regions with low-energy localized states along transport direction[43], which is corresponding to the

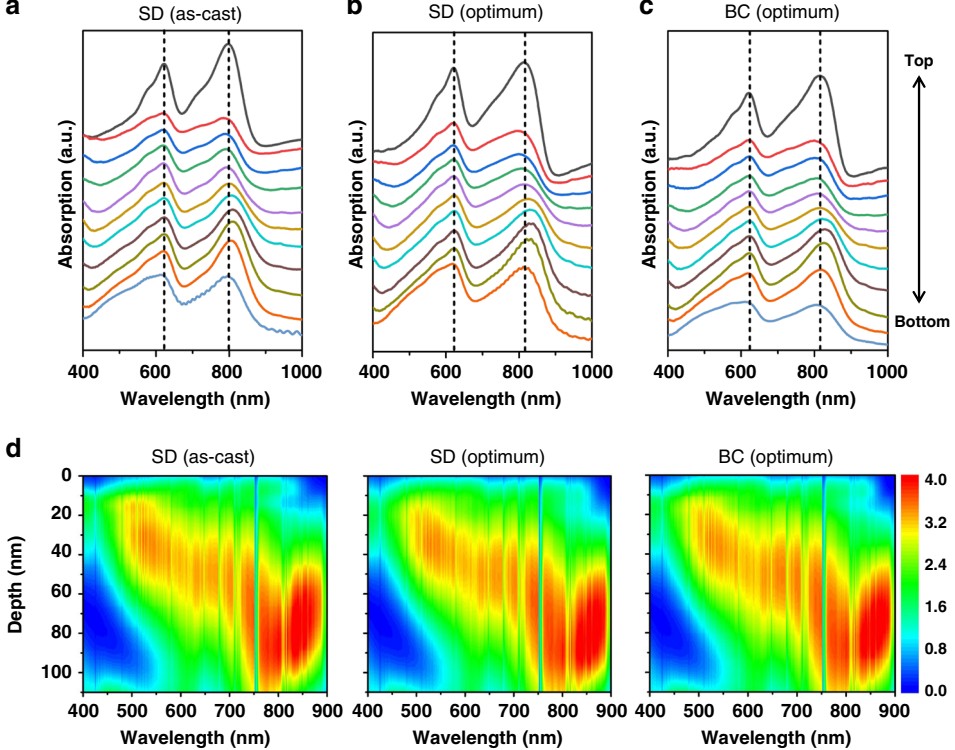

**Fig. 2 Film-depth-dependent absorption spectroscopy and exciton generation. a** SD (as-cast) film. **b** SD (optimum) film. **c** BC (optimum) film. The spectra were vertically shifted and rescaled for clarity. The dashed vertical lines show the absorption peaks of PT2 and Y6. The sub-layer thickness for each spectrum is approximately 10 nm. **d** Calculated exciton generation contours (unit, $10^{25}$ $m^{-3}\cdot s^{-1}\cdot nm^{-1}$ at its film-depth and wavelength) from **a** to **c**, respectively. The noise-like vertical lines in the contours are from the real fluctuation of solar spectra (AM 1.5 G). 0 nm and 110 nm represent active layer/PNDIT-F3N-Br and PEDOT:PSS/active layer interfaces, respectively.

fact that PT2 is a high-performance donor material and its fibril network represents an ideal framework for bulk heterojunction. On the other hand, as compared with the as-cast SD films, the absorption peaks of Y6 in the optimum SD and BC films show a red-shift by 20 nm, demonstrating a higher crystalline ordering of Y6 molecules in the blends. Moreover, this crystalline improvement occurs at every film depth, contributing to the increased electron mobility.

Subsequently, we numerically simulated the exciton distribution, using the film-depth-dependent light absorption spectra in combination with optical interferences among the different layers within the actual photovoltaic devices. The simulation is performed upon a modified transfer-matrix optical method[45]. Figure 2d shows that the excitons are generated within the entire donor:acceptor active layer, and the population maximizes in the middle regions of the layer (Fig. 2 and Supplementary Fig. 11). Nevertheless, exciton generations within PT2 and Y6 depend on film depth in different manners. As shown by the exciton generation contour (Fig. 2d), the excitons generated in PT2 phase (wavelength around 600 nm) are mainly located in the top part of the active layer. This means that the holes after dissociation of such excitons need to transport across tens of nanometers of the bottom part of the active layer toward anodes. The high quality PT2 fibril framework at the bottom part of the film provides an ideal pathway for the collection of holes by anode. On the other hand, most of the photons absorbed by Y6, which corresponds to wavelength 700–900 nm, generates excitons in the bottom part of the active layer. Therefore, morphologically continuous Y6 network within PT2 network warrants the transport of exciton-dissociation-induced electrons toward cathode. Interestingly, as shown by Fig. 1d, in the wavelength range between 800 nm and 900 nm, the EQE of optimum SD device is much higher than that

of as-cast SD device, which demonstrates a highly efficient exciton dissociation and subsequent high-speed charge transport toward respective electrodes. Consequently, both EQE profile and exciton generation contour prove a good interpenetrated network in PT2:Y6 layer.

The film morphology is studied by atomic force microscopy (AFM) measurements. As shown in Fig. 3 and Supplementary Fig. 12, neat PT2 and SD films showed obvious nanofiber structures, suggesting that PT2 nanofibers distribute on the top of the films, well consistent with the analysis of film-depth-dependent light absorption spectra. BC film also shows well distributed interpenetrating network morphology. Such a nanofiber morphology can offer an effective hole transport channel.[36] In detailed comparison, the as-cast and optimum SD films exhibited root mean square (rms) values of 2.31, and 2.53 nm, respectively, indicating that the optimum SD film has a relatively rough surface. This is because the DIO additive could increase the crystallinity of PT2 and surface roughness of the blend film.

Grazing incidence wide-angle X-ray scattering (GIWAXS) and resonant soft X-ray scattering (RSoXS) are performed to investigate the film morphology. 2D GIWAXS diffraction patterns and the corresponding line-cut profiles of neat films are shown in Supplementary Fig. 13a–c. It can be seen that PT2 and Y6 both exhibit strong (010) diffraction peaks at $q_z$ = 1.64 $Å^{-1}$ and 1.78 $Å^{-1}$, in the out-of-plane direction, corresponding to π-π stacking distances of 3.83 nm and 3.53 nm, respectively, indicating the preferred face-on orientation of molecule ordering in both neat films. GIWAXS results of SD and BC thin films were shown in Fig. 4a–c. Similar peak positions were recorded for these samples. The merging in lamellar ordering and π-π stacking prohibited detailed peak fitting and thus combined intensity was used to estimate the overall ordering

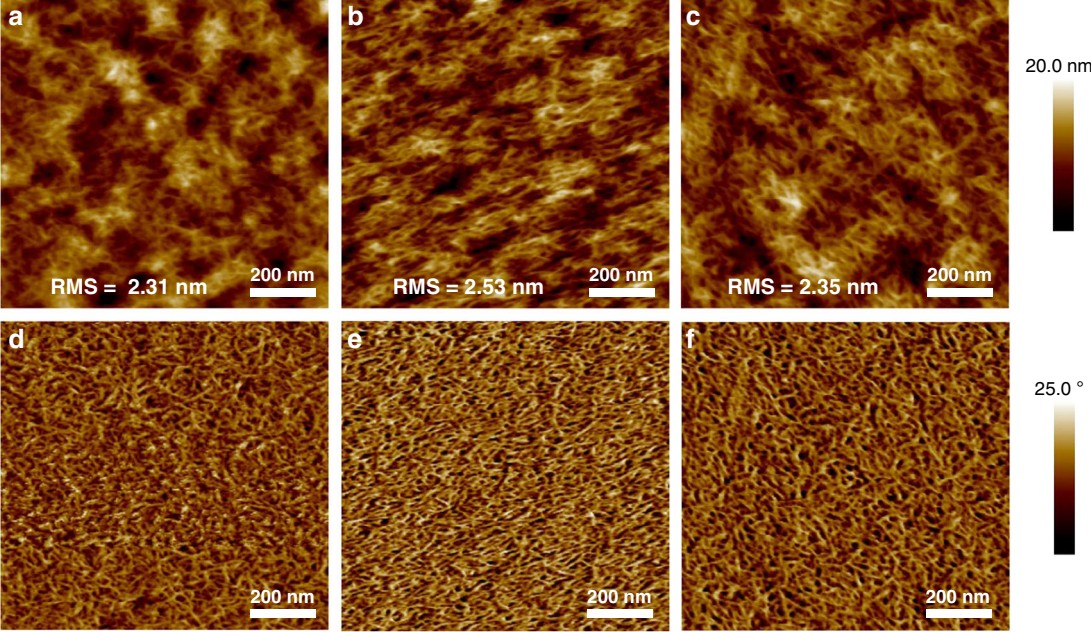

**Fig. 3 AFM topography images of SD and BC blend films. a, d** SD (as-cast) film. **b, e** SD (optimum) film. **c, f** BC (optimum) film.

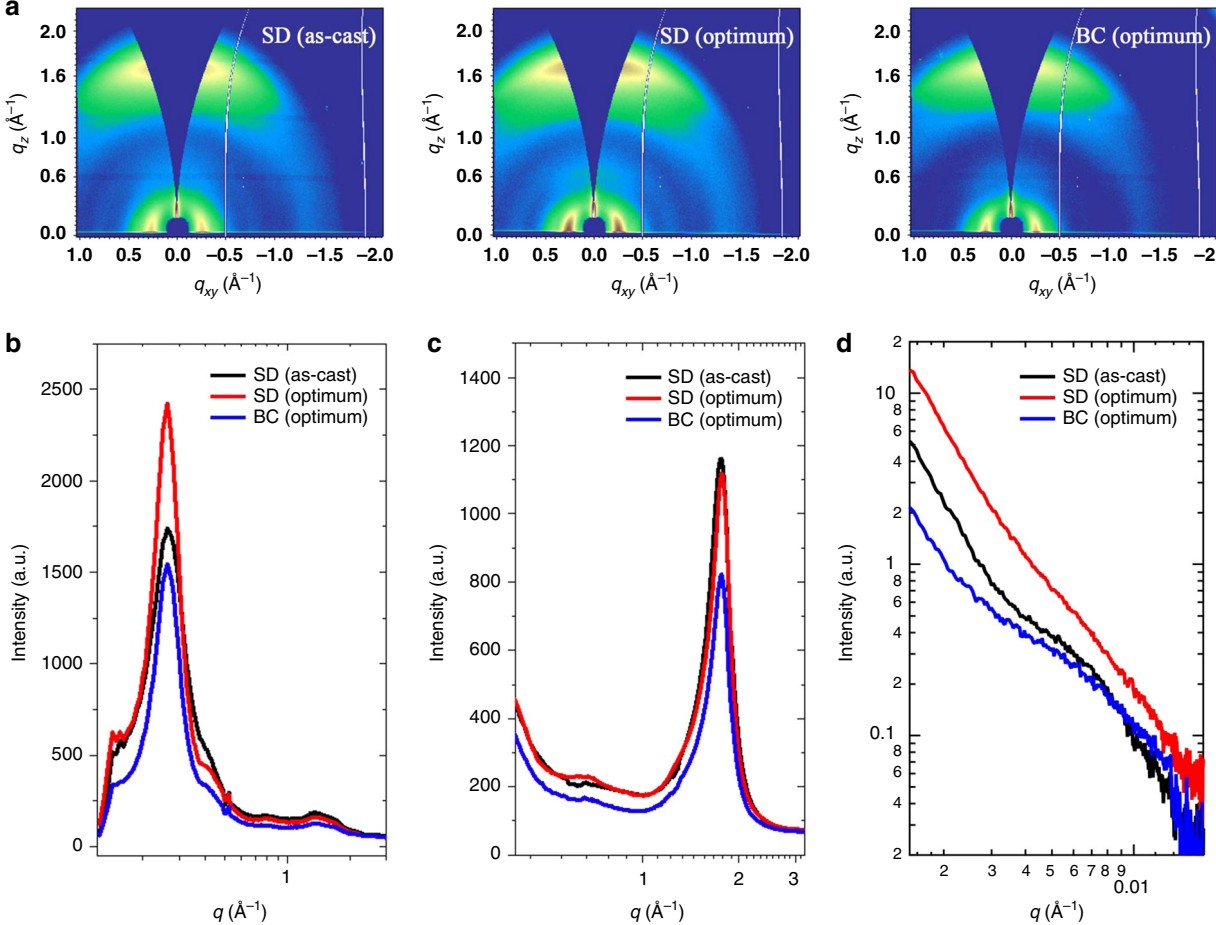

**Fig. 4 Morphology characterizations of SD and BC blend films. a** 2D GIWAXS diffraction patterns of SD and BC blend films. **b** In-plane (black lines) and **c** out-of-plane (red lines) line-cut profiles of GIWAXS diffraction patterns. **d** RSoXS scattering profiles of the SD and BC blend films using a photon energy of 284.2 eV.

of different thin films. As-cast SD thin film showed a relatively low lamellar ordering as shown in in-plane diffraction intensities in $0.26\,Å^{-1}$. When DIO was used, such intensity was largely increased. Thus the presence of DIO recovered the lamellar ordering in PT2 phase, forming better-defined fibril structure. The π-π stacking at $1.75\,Å^{-1}$ showed similar intensities for as-cast SD and optimum thin films, indicating the anisotropic function in regulating structure ordering of DIO content. It should be noted that optimum BC thin film showed moderate lamellar and π-π ordering in GIWAXS, and thus the crystallinity in such process is relatively lower when compared with the optimized SD film, leading to reduced transport and device efficiency. These results revealed fundamental advantage of SD processing, which could retain a good polymer crystalline framework and high crystallinity upon optimum SD methods. Improved polymer crystallization reduces amorphous content that could reduce the polymer-NFA interactions that leads to complicated morphology. Thus the morphology formation complexity was transformed to the crystalline fiber framework construction issue, which is much easier to handle, as seen from our results. It would be beneficial to synthesizing donor polymers with extreme narrow polydispersity index (PDI) to fully investigated the detailed polymer and NFA interaction, which, however, is limited by chemical synthesis difficulties and thus abandoned.

The phase separation of the blend films was studied by using RSoXS. As presented in Fig. 4d, the as-cast SD PT2:Y6 film showed a scattering peak at $q = 0.0071\,Å^{-1}$, which corresponded to a distance of 88.4 nm. The optimum SD film that used DIO and thermal annealing treatment showed enhanced RSoXS intensity indicating a more rigorous phase separation, which corresponded to a better-defined PT2 fiber network structure as indicated by GIWAXS characterizations. As for RSoXS profiles of the optimum BC blend, a similar phase separation size scale was seen comparing to SD thin film, yet the intensity was low, comparable to as-cast SD thin film. Such result is ascribed to the blended solution casting, in which the interaction of PT2 and Y6 retards the crystallization of each other, leading to better mixing in condensed film. And thus lower crystallinity was seen in GIWAXS characterization. GIWAXS and RSoXS indicated important features in SD and BC thin film preparations, in which the volatile solvent casting of Y6 could partially disturb the ordering of PT2 first layer and introduce Y6 content inside to form BHJ morphology. The presence of DIO and subsequent thermal annealing treatment could effectively recover the PT2 ordering, leading to better-defined fibril network morphology that comparable, if not better than, the BC prepared BHJ thin film.

**Sequential-deposition OSCs by Slot-die printing.** Spin coating is commonly used to fabricate OSCs on a laboratory scale, while for industrial processes, blade coating or slot-die printing processes are the standard processes. Herein, we use slot-die printing to fabricate the SD OSCs. The donor and acceptor solutions were firstly pumped into the slot-die head and then deposited onto a substrate sequentially, with slot-die running at a constant speed to coat the film. The coating thickness is dictated by solution concentration, substrate translated rate, and distance between the head and substrate. As shown in Supplementary Fig. 14 and Table 5, the optimum SD OSCs demonstrate excellent photovoltaic performance with a PCE of 15.5% at a device area of $0.04\,cm^2$. Moreover, the large-scale devices ($0.8\,cm^2$) are also fabricated and an impressively high PCE of 14.6% is achieved. In contrast, slot-die printed BC OSCs showed much lower PCEs of 14.5% and 12.6% with device area of $0.04\,cm^2$ and $0.8\,cm^2$,

respectively. This result demonstrates the advantage of SD method that is more robust in morphology control and scalable in large-area device fabrication.

**Sequential deposition and polymer batch tolerance.** The SD device fabrication disentangles the complicated donor-acceptor interactions as seen in BC processing. Thus the underlayer morphology can be well developed. PT2 neat thin film can form nanofibers that enhances charge transport. Such nanostructure formation is readily controllable when molecular weight reaches certain value to facilitate interchain aligned crystallization. And thus SD processing can be more robust in morphology control when polymer batch-to-batch variation is considered. PT2 with different molecular weight ($M_n$) ranging from 45 kDa to 91 kDa were synthesized and used to fabricate OSCs by SD and spin coating methods. As presented in Fig. 5a, b and Table 2, the SD OSCs showed performance insensitivity to the PT2 batch variations, much better in statistics comparing to BC OSCs. The SD OSCs yielded high PCEs of not less than 16% in three batches of PT2. As for BC OSCs, the higher molecular weight of PT2 leads to lower PCEs (chloroform as the blend host solvent). For eliminating the influence of casting solvent on the component solubility, we also fabricated the BC OSCs where the blend was casted form chlorobenzene, which also exhibited an inferior device performance as shown in Supplementary Fig. 15 and Table 6. In order to investigate if the secondary solution-casting can mess with the preformed network, we preformed the AFM measurements. As presented in Supplementary Fig. 16, all the PT2 polymers with different molecular weight show clear nanofiber structure. After CF/DIO solvent washing, such nanofiber structure was remained well for all the polymers, indicating that the secondary solution-casting does not mess with the preformed network. In addition, the photovoltaic characteristics of PT2 films with CF/DIO solvent washing of fibril layer have been also investigated. As shown in Supplementary Fig. 17 and Table 7, the photovoltaic characteristics of PT2 films with CF/DIO solvent washing exhibited quite small difference to those without solvent washing.

Similar results were also obtained when another efficient small molecular acceptor IT-4F was applied. As shown in Fig. 5c, d, the SD OSCs yielded high PCEs of near 13%. However, the BC OSCs shows inferior PCE when the molecular weight of PT2 was increased (chlorobenzene as the blend host solvent). Moreover, we used the chloroform as blend solvent to fabricate the BC OSCs. As shown in Supplementary Fig. 18 and Table 8, the performance of BC OSCs shows a similar trend that the device performance decreased with the increase of PT2 molecular weight. To further verify the results, we randomly synthesized two batches of PT2 by controlling the polymerization time. As shown in Supplementary Fig. 19 and Table 9, when blended with Y6, the SD OSCs still yielded PCEs of more than 16%, which is much higher than that of BC OSCs. We employed the photo-CELIV to investigate the carrier mobility of the different PT2 batches in the PT2:Y6 system. As presented in Supplementary Fig. 20 and Table 10, the carrier mobility of SD OSCs is twice of the corresponding BC OSCs, suggested that the high charge transport channel is maintained in the SD films of different PT2 batches. Such results are quite encouraging, revealing the possibility of using SD to eliminate the polymer synthesis issues in OSC applications.

The AFM topography images of SD and BC films processed with different PT2 batches were also studied. As shown in Supplementary Fig. 21, SD films show obvious and similar nanofiber structures with RMS values of around 2.5 nm. As for BC films (casted from chloroform), the surface morphology based

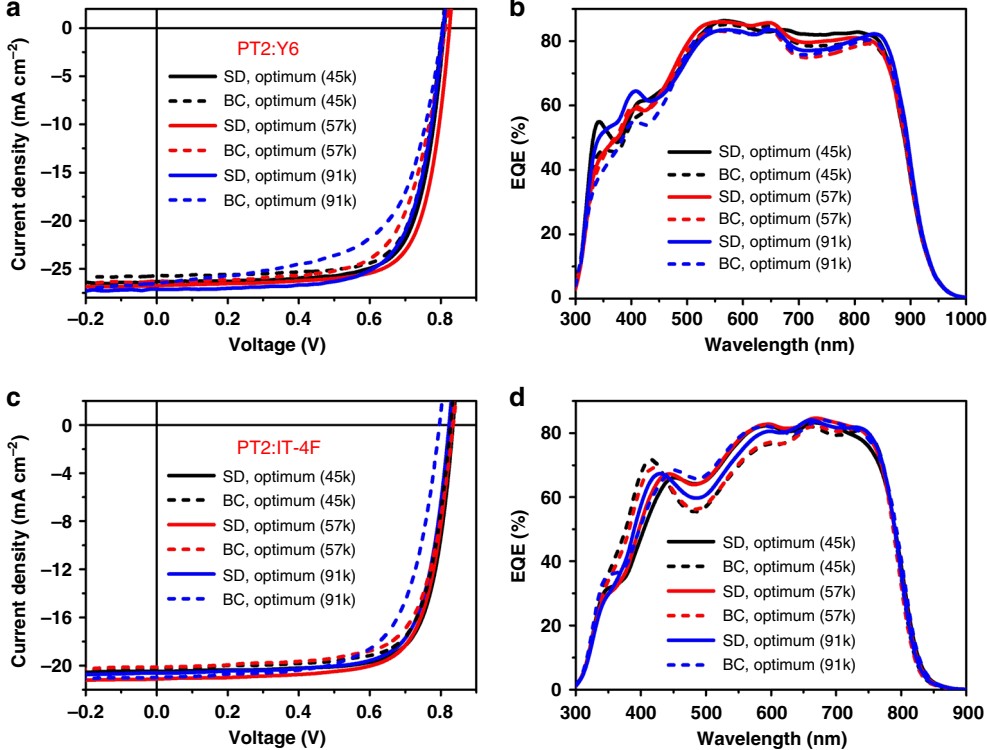

**Fig. 5 Photovoltaic performance of OSCs based on different PT2 batches. a,c** J–V characteristics of SD and BC OSCs fabricated with different PT2 batches under constant incident light intensity (AM 1.5 G, 100 mW cm$^{-2}$), and **b,d** the corresponding EQE spectra.

| Table 2 Summary of device parameters of SD and BC OSCs. | | | | | | |
| --- | --- | --- | --- | --- | --- | --- |
| Blend | Molecular weight (M$_n$) | Operating conditions | V$_{oc}$ (V) | J$_{sc}$ (mA cm$^{-2}$) | FF (%) | PCE (%) (average)$^a$ |
| PT2:Y6 | 45 kDa | SD | 0.82 | 26.3 (25.8)$^b$ | 74.7 | 16.1 (16.0 ± 0.1) |
| | | BC | 0.82 | 25.7 (25.3) | 74.1 | 15.6 (15.3 ± 0.3) |
| | 57 kDa | SD | 0.83 | 26.7 (26.2) | 74.4 | 16.5 (16.3 ± 0.2) |
| | | BC | 0.83 | 26.3 (25.6) | 68.9 | 15.0 (14.8 ± 0.2) |
| | 91 kDa | SD | 0.82 | 27.1 (26.4) | 71.8 | 16.0 (15.9 ± 0.2) |
| | | BC | 0.81 | 26.5 (26.0) | 62.4 | 13.4 (13.2 ± 0.1) |
| PT2:IT-4F | 45 kDa | SD | 0.84 | 20.5 (20.1) | 75.7 | 13.0 (12.8 ± 0.2) |
| | | BC | 0.83 | 20.2 (19.5) | 74.5 | 12.5 (12.3 ± 0.3) |
| | 57 kDa | SD | 0.83 | 21.1 (20.3) | 73.9 | 12.9 (12.8 ± 0.1) |
| | | BC | 0.84 | 20.1 (19.4) | 71.5 | 12.1 (11.7 ± 0.4) |
| | 91 kDa | SD | 0.83 | 20.6 (20.0) | 74.2 | 12.7 (12.6 ± 0.2) |
| | | BC | 0.80 | 21.0 (20.5) | 67.6 | 11.4 (11.1 ± 0.4) |
| $^a$The average parameters were calculated from 10 independent cells. | | | | | | |

on three PT2 batches shows huge difference, and the RMS value of the PT2-based films (M$_n$ = 91 kDa) increased to 3.52 nm (Supplementary Fig. 22). Such rough surface caused by the strong aggregation is negative for the OSCs performance. The same surface morphology variation was also occurred in the PT2:IT-4F systems as shown in Supplementary Figs. 23 and 24.

## Discussion

In this manuscript, we demonstrate a two-step sequential deposition method for the formation of the optimum active layer morphology to achieve high-efficiency OSCs. The success of the fabrication lies in the combination of FNS and SD. PT2 copolymer that self-assembles into fibril structure guarantees a good morphology framework and good carrier transport. The utilization of a small amount of DIO additive in the second layer deposition is proven to be critical, which increases the

crystallinity of PT2 due to the slow evaporation induced crystallization, a process similar to solvent vapor annealing treatment. A high PCE of 16.5% is achieved in the PT2/Y6 material combination. Besides the high efficiency recorded, other distinctive advantages are seen. The readily scaling up capability makes the FNS-SD method a very promising way in organic photovoltaic (OPV) panel fabrication. The robust morphology, taking advantage of the fibril network crystalline structure, could eliminate polymer batch-to-batch variation, which is unexpected and highly promising in OPV research.

## Methods

**Materials**. The synthetic procedures of PT2 is similar to that of most random tepolymers[46] and the detailed synthetic procedures and characterizations of PT2 were provided in Supplementary Methods and Supplementary Figs. 1–3. Y6 was purchased from Hyper Inc. PNDIT-F3N-Br was synthesized in our group according to the reference[47].

**OSC device fabrication and measurement**. The OSCs are fabricated with a conventional structure of ITO/PEDOT:PSS/active layer/PNDIT-F3N-Br/Ag. The ITO substrates are cleaned by ultrasonic for 15 min with detergent, distilled water, acetone and *iso*-propanol in sequence, then transferred to drying oven and staying overnight. After that, the ITO substrates are subjected to UV-ozone treatments for 15 min, followed by spin-coating with PEDOT:PSS solution that filtered by 0.45 mm poly(tetrafluoroethylene) (PTFE) filter at 4000 rpm for 30 s. Then the ITO substrates are baked at 150 °C for 15 min and transferred to a nitrogen-filled gloved box. As for the fabrication of SD OSCs, PT2 in chlorobenzene solution is spin-coated on the top of PEDOT:PSS film, and then Y6 (or IT-4F) dissolved in chloroform solution is deposited atop PT2 film by spin-coating. As for PT2:Y6 BC OSCs, the blend solution (PT2:Y6 = 1:1.2 wt%) in chloroform is spin-coated on the top of the PEDOT:PSS layer at different speeds. For BC OSCs based on PT2:IT-4F, the blend solution (PT2:IT-4F = 1:1 wt%) in chlorobenzene is spin-coated on the top of the PEDOT:PSS layer. After thermal annealing for 10 min, a methanol solution of PNDIT-F3N-Br at a concentration of 0.5 mg/mL is spin-coated onto the active layer at 3000 rpm for 30 s. Finally, silver (100 nm) is thermally evaporated through shadow masks with a device area of 4 mm$^2$. The current-voltage characteristics of solar cells are measured by Keithley 2400. Solar cell performance used an Air Mass 1.5 Global (AM 1.5 G) solar simulator (SS-F5-3A, Enlitech) with an irradiation intensity of 100 mW cm$^{-2}$, which was measured by a calibrated silicon solar cell (SRC2020, Enlitech). The $J$–$V$ curves are measured along the forward scan direction from −0.5 to 1 V, with a scan step of 50 mV and a dwell time is 10 ms. The external quantum efficiency (EQE) spectrum of OSCs are recorded using Enli QE-R3011 (Enli Technology Co., Ltd Taiwan).

**SCLC measurements and film thickness**. The SCLC is measured with hole-only (ITO/PEDOT:PSS/active layer/MoO$_3$/Ag) and electron-only (ITO/ZnO/active layer/PNDIT-F3N-Br/Ag) devices under dark condition. The mobility is determined by fitting the quadratic SCLC region, which is described by the equation, $J = (9/8)\varepsilon_0\varepsilon_r\mu_0 V^2/L^3$, where $\mu_0$ is the zero-filed mobility, and $\varepsilon_0$ and $\varepsilon_r$ represent the permittivity of free space and relative permittivity, $V$ represents the effective voltage and $L$ is the thickness of the active layer, respectively. Active layer thickness is obtained by a surface profiling system (DektakXT, Bruker). The samples are prepared on ITO/PEDOT:PSS substrate.

**Photo-CELIV measurement**. Photo-CELIV measurement was performed by the all-in-one characterization platform, Paios (Fluxim AG, Switzerland). The device structure is ITO/PEDOT:PSS/active layer/PNDIT-F3N-Br/Ag.

**Film-depth-dependent light absorption spectra**. Film-depth-dependent light absorption spectroscopy was conducted by a home-made setup. Low-pressure (less than 20 Pa) oxygen plasma was used for the incremental etching of the film. The UV–Vis absorption spectrum after each etching was monitored by an optical spectrometer. Beer-Lambert's law was utilized to fit the film-depth-dependent light absorption spectra. The detailed methods about the measurements and numerical fitting were available elsewhere[48].

**GIWAXS**. The GIWAXS characterization of the thin films is measured on beamline 7.3.3 at the Advanced Light Source (Lawrence Berkeley National Laboratory). All samples are prepared under device conditions on the silicon wafer substrate. The scattering signal of samples is recorded with a pixel size of 0.172 mm by 0.172 mm (Pilatus 2 M detector). The distance between the samples and beam center is approximately 300 mm which calibrated by the silver behenate standard. The incidence angle is set to be 0.16°. The beam energy is 10 keV, operating in top-off mode. A 30 s exposure time on a 2D charge-coupled device (CCD) detectoris recorded to collect the diffraction signals. All GIWAXS measurements are done in a helium atmosphere.

**RSoXS**. The RSoXS is measured at beamline 11.0.1.2 at Advanced Light Source, Lawrence Berkeley National Laboratory. All samples are prepared under device conditions on the Si/PEDOT:PSS substrates. The blend films are then floated in water and transferred to a silicon nitride window. The scattering signals are collected in a vacuum by using a Princeton Instrument PI-MTE CCD (charge-coupled device) camera.

**Reporting summary**. Further information on research design is available in the Nature Research Reporting Summary linked to this article.

## Data availability
The experiment data that support the findings of this study are available from the corresponding author upon reasonable request. The source data underlying Supplementary Fig. 6 and Tables 1, 2, as well as Supplementary Tables 1–9 are provided as a Source Data file. Source data are provided with this paper.

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

## Acknowledgements

This work was financially supported by the National Natural Science Foundation of China (NSFC) (Grant Nos. 51825301, 21734001 and 21674007). X-ray data were acquired at beam lines 7.3.3 and 11.0.1.2 at the Advanced Light Source, Lawrence Berkeley National Laboratory, which is supported by the Director, Office of Science, Office of Basic Energy Sciences, of the U.S. Department of Energy under Contract No. DE-AC02-05CH11231. Y.S. gratefully acknowledges Prof. Jianhui Hou (ICCAS) for the assistance with photo-CELIV measurements.

## Author contributions

K.W. fabricated and characterized the devices. L.Y. synthesized the PT2 polymer. L.Z., J. X., J.Z. and F.L. performed the morphology characterization and analysis; F.X. and G.L. measured the film-depth-dependent light absorption spectra and performed the optical simulation; S.T. participated in the discussion of PT2 synthesis; F.L. and Y.S. supervised and directed this project; K.W., F.L. and Y.S. wrote the manuscript. All authors commented on the manuscript.

## Competing interests

The authors declare no competing interests.
