## [Peer Review File · Nature Communications]

Reviewers' Comments:

Reviewer #1:

Remarks to the Author:

The authors reported a sequential deposition method of organic layers for organic photovoltaic devices, claiming that the method is desirable for achieving a reliable scale-up solution process. The formation of a crystalline fibrillar network structure of the donor (PT2) material followed by solution-phase diffusion of the acceptor material (Y6) looks logically straightforward and seems valid in eliminating kinetically-driven morphology variation problems. The resultant high device performance is promisingly for larger-scale productions. However, the novelty of this work is somewhat limited considering that the use of fibril structure to make a robust and reliable morphology has been widely reported within the last 10 years, not only with the common P3AT series but also in D-A type polymers. Simply replacing the materials with champion materials would be inadequate to meet the high standard of Nature Communications. Furthermore, there are several technical issues to be addressed.

Consequently, the reviewer suggest a major revision of the entire manuscript before it can get published in highly-esteemed journals such as Nature Communications.

Technical issues

- The authors first claimed the robustness of the network structure in the SD processing. However, the reviewer finds it unconvincing since the authors have not shown any direct evidence that the secondary solution-casting does not mess with the preformed network. The fact that polymer above certain MW all form fibril network (which they also did not show any data) is not enough to prove their point.
- Particularly, the authors themselves mentioned that the use of CF solvent quickly dissolves the amorphous region of the network and it can be beneficial in achieving best morphology. Since the basic traits of polymers such as PDI and MW diverge the solubility of the polymer greatly, it is reasonable to deduce from their own findings that the use of SD method does not necessarily guarantees the reliable and reproducible production irrespective of batch-to-batch variations.
- Expanding on that, AFM, GIWAXS, and RSoXS can be indirect evidence for the formation of more defined morphologies, but they only prove that SD approach results in better morphology for PT2 and Y6 blends and does not support their claims on general processing innovations. More systematic studies, such as the analyses of morphological and photovoltaic characteristics with CF (or CF/DIO) solvent washing of fibril layer comprising polymers with various PDIs and MWs, would be necessary.
- It is questionable that the homogeneous vertical phase distribution (unlike inhomogeneous ones in other SD method works) is derived only from the fibril-based SD processing. The authors did not explain enough about Fig 2c in their difference with Fig 2a and b, which can show that whether these characteristics are from molecular properties or processing methods.
- In order to claim the innovative aspects of the work, it might be beneficial to show how a gradual scaling-up of processing deviates the morphological development in both SD and BC processing.

Reviewer #2:

Remarks to the Author:

The solution processing sequential deposition (SD) method has been paid attention owing to its advantages such as large-area printing and ease of film morphology control with high reproducibility. As aforementioned merits of SD, herein, they demonstrated the facilitation of fibril network strategy in the donor polymer layer within the SD films. For the depth analysis in the vertical direction as one of the crucial investigations of SD film, in addition, the film-depth-dependent light absorption spectroscopy clearly revealed the donor and acceptor composition of SD films, as well as exciton generation and dissociation within the photo-active layer. Moreover, the 0.8 cm² of large-area fabrication via slot-die printing with high PCE up to 14.6% ensures the worth of the SD system they found. Although the characterizations of their synthesized polymer

PT2 are somewhat insufficient, they did a well-organized study about the SD system and the reviewer would recommend the publication of this paper after following minor revisions.

1. The used PT2 polymer is not only random terpolymer of PBT1-C (Adv. Mater., 2018, 30, 1803045) and PBT1-C-2Cl (J. Mater. Chem. A, 2019, 7, 8055), but also firstly reported in this article. In the case of brand-new material, the full characterization data be always required. However, there is no characterization synthesized PT2 polymer such as NMR spectra, elemental analysis and so on. Especially, the molecular frontier orbitals of PT2 is not provided, but it might be changed compared to PBT1-C and PBT1-C-2Cl. They should properly characterized the new polymers like the papers (Nat. Commun. 2018, 1867). Besides, the authors need to provide some texts related on the random terpolymers (see the random review paper DOI:10.1002/adma.201804762)

2. They showed successful synthesis of different molecular weight of PT2 polymers and investigated the effects of molecular weight on the SD system. However, they didn't describe how to they control the molecular weight clearly. I recommended the author to supplementary describe the synthesis of polymer including aforementioned point 1.

Reviewer #3:

Remarks to the Author:

The authors describe the use of a conjugated polymer (PT2) and a small molecule acceptor (Y6) to reach very high power conversion efficiencies (16.5%) via a sequential deposition (SD) in which PT2 is deposited, followed by Y6. The main claim of the authors is that the SD method is advantageous compared to blend coating (BC) in which the PT2:Y6 mixture is deposited in one step.

The authors have submitted another paper (ref. 40) which describes, amongst others, PT2:Y6 blends for solar cells reaching a maximum efficiency of (15.6%). The efficiencies are similar and hence the main question to be answered is if the SD method is really advantageous compared to BC. There are already several reports on sequential deposition of semiconductor blends for organic solar cells, some are cited in the manuscript. The main advantage of a two-step procedure would be a significantly better performance or the fact that in SD less hazardous solvents could be used. In this example that is not the case. PT2 is deposited from chlorobenzene, followed by Y6 from chloroform (CF) with diiodooctane (DIO) as co-solvent. Then the layer is thermally annealed at 100 °C. In BC the blend is deposited from CF/DIO and also thermally annealed.

The authors argue in the abstract that the two-step SD eliminates the kinetic issue in morphology formation. But that is not what the data show. In fact in the manuscript the authors assume a certain amount of PT2 fibrils is dissolved during the second layer casting and demonstrates that presence of DIO is needed to recover the PT2 fibril structure. It is difficult to see the advantage of a two-step procedure if the second step is identical to the procedure that would be used in a one-step (BC) deposition. There are indeed small differences between SD and BC deposited films, but nothing dramatic.

Some other smaller issues:

1. Line139: 3.152 cm² should be 3.152 mm²

2. Line 142: they reach 15.5% in ref 40 for the BC device, now 15.0%, why is it less?

3. Line 162: The authors present a good stability for the SD OSC, but there is no comparison with BC OSC devices. Why would it be an advantage?

4. The discussion on the hole and electron mobilities show that all are of the same magnitude. The SCLC method is not very accurate and the three significant digits given are not realistic. If they want to maintain their numbers, the authors should study the SCLC behavior at different thickness, prove the L-3 behavior and show that the mobility is constants. Now they fit a very small part of J-V curve for an unknown thickness where by chance the current density scales with square of the voltage. I know it is done in many papers, but it is not convincing. Supp. Fig 5, lacks a vertical scale. The current density of the BC electron-only device is higher than that of the BC device but the mobility is less.

5. Concerning the ideality factor the optimum SD device showed a slope of 1.25 kT/q , while the optimum BC device has slope of 1.41 kT/q . The authors conclude that the SD method suppresses the trap-assisted recombination in the blends compared to the BC method. This is a too simple line of a reasoning. First the V_{oc} of the two devices is the same, hence in terms of non-radiative recombination, the SD is not better than the BC. The authors forget that an ideality factor 1 is not only expected from bimolecular recombination, but also for surface recombination. The ideality factor itself is not a very meaningful parameter in practice.

Reviewer #1

The authors reported a sequential deposition method of organic layers for organic photovoltaic devices, claiming that the method is desirable for achieving a reliable scale-up solution process. The formation of a crystalline fibrillar network structure of the donor (PT2) material followed by solution-phase diffusion of the acceptor material (Y6) looks logically straightforward and seems valid in eliminating kinetically-driven morphology variation problems. The resultant high device performance is promisingly for larger-scale productions. However, the novelty of this work is somewhat limited considering that the use of fibril structure to make a robust and reliable morphology has been widely reported within the last 10 years, not only with the common P3AT series but also in D-A type polymers. Simply replacing the materials with champion materials would be inadequate to meet the high standard of Nature Communications. Furthermore, there are several technical issues to be addressed.

Answer: Thanks for the comments. We understand the concern from the reviewer arguing the novelty of the manuscript. It is true that layer-by-layer (LBL) approach has been reported years ago regarding the OPV fabrication. However, the current research bears a much broader horizon that targets the very essence of high efficiency NFA large area device fabrication. We have tried direct printing of solar cells based on Y6 blends, yet with poor luck to make high performance devices. We analyzed the electronic and morphological issues behind, and then developed the sequential casting method, which turned out to be quite successful. We tried our best effort to grasp the scientific content of the SD method in NFA OPVs to avoid being simple incremental device optimization, and do made interesting findings that fundamentally different from previous layer-by-layer fabrication reports. Important findings are summarized as follows: (1) A well characterized morphology showing unexpected homogeneous vertical segregation is seen, which is quite different from previous LBL reports. Such results obtained from the success of highly quality fibril networks. And we developed detailed fabrication procedures of reconstructing fibril network upon second layer preparation. These features in combination with thin film physical property correlation would be of high

interest to OPV community, which paved way for high efficiency large area device printing. (2) The establishing of the first layer fibrillar morphology and effective reconstruction second layer fabrication bring in very important merit that could separate the complicated material crystallization and interaction, which opens a new approach in morphology optimization and fundamentally changes the difficulty of polymer batch-to-batch variation issue. While different polymer batch blend with NFA in spin casting could lead to different morphology and performances, the layer-by-layer method, of making a fibril framework first underneath is of particular advantage. Although the second layer casting can partially disrupt the first layer morphology, a DIO repairing mechanism could solve the issue easily. Thus no polymer batch-to-batch variations are seen, which we think is a fundamental breakthrough in OPV. The findings in basic science and high performance in a scalable fabrication make the current manuscript quite unique amount OPV publications, which deserve attention in the community. The previous submission many not be that strong in supporting these arguments. We added much more data in revised version to highlight the importance of SD fabrication.

(1) The authors first claimed the robustness of the network structure in the SD processing. However, the reviewer finds it unconvincing since the authors have not shown any direct evidence that the secondary solution-casting does not mess with the preformed network. The fact that polymer above certain MW all form fibril network (which they also did not show any data) is not enough to prove their point.

Answer: Thanks for the comments. The polymers used in research are prone to form fibrils due to high co-facial π - π stacking. The fibrillar type of crystallization are high in energy landscaping comparing to material interactions, which is the fundamentally advantage. In order to investigate if the secondary solution-casting can mess with the preformed network, we preformed the AFM measurements. As presented in Fig. R1, all the PT2 polymers with different molecular weight show clear nanofiber structure. After CF/DIO solvent washing, such nanofiber structure was remained well for all the

polymers, indicating that the secondary solution-casting does not mess with the preformed network.

In addition, the photovoltaic characteristics of PT2 films without and with CF/DIO solvent washing of fibril layer have been also investigated. As shown in Fig. R2 and Table R1, the photovoltaic characteristics of PT2 films without CF/DIO solvent washing exhibited quite small difference to those with solvent washing.

Fig. R1 AFM phase images. **a,d** PT2 film ($M_n=45K$) without and with CF/DIO washing. **b,e** PT2 film ($M_n=57K$) without and with CF/DIO washing. **c,f** PT2 film ($M_n=91K$) without and with CF/DIO washing.

Fig. R2 Photovoltaic performance. **a** $J-V$ characteristics of SD OSCs fabricated with different PT2 batches under constant incident light intensity (AM 1.5G, 100 mW cm^{-2}),

and **b** the corresponding EQE spectra. The PT2 sublayer was washed with CF/DIO solvent (1% DIO).

Table R1 Summary of device parameters of the SD OSCs fabricated with PT2 films with CF/DIO solvent washing.

Blend	M_n and PDI	V_{oc} (V)	J_{sc} (mA/cm ²)	FF (%)	PCE (%) (average) ^a
PT2:Y6	45k (2.33)	0.83	25.8 (25.6) ^b	75.1	16.1 (15.9±0.2)
	57k (2.05)	0.82	26.2 (25.9)	74.5	16.3 (16.2±0.1)
	91k (2.08)	0.82	26.4 (26.1)	74.0	16.0 (15.8±0.2)

^a The average parameters were calculated from 10 independent cells.
^bEQE values.

(2) Particularly, the authors themselves mentioned that the use of CF solvent quickly dissolves the amorphous region of the network and it can be beneficial in achieving best morphology. Since the basic traits of polymers such as PDI and MW diverge the solubility of the polymer greatly, it is reasonable to deduce from their own findings that the use of SD method does not necessarily guarantees the reliable and reproducible production irrespective of batch-to-batch variations.

Answer: Thanks for the comments. As the reviewer mentioned, there is solubility difference for the polymers with different PDI and M_n , which can cause significant performance deviation for OSCs fabricated using spin coating. In SD casting, we actually need to look into polymers from a different perspective. The first layer is a fibril network embedded in amorphous matrix. The second layer casting is a nonequilibrium process that the dissolution time towards crystalline fibril and amorphous matrix is quite different. Conjugated polymers are of rigid backbone nature, and chain length variation regarding to physical property change is more pronounced in low molecular weight region (usually below 10K M_n in P3HT experience). Once outreach the molecular weight, the physical properties will be quite similar in homogenous thin films. The current work actually shifts the PDI and M_n issue to a thin

film crystallinity and nonequilibrium solubility issue, which are much easier to handle and could easily reproduced. We demonstrate these characteristics with much more robust data crossing different BHJ blends in the revised version. The SD OSCs based on PT2:Y6 showed performance insensitivity to the PT2 batch variations, which is much better in statistics comparing to BC OSCs. We tried another efficient small molecular acceptor (IT-4F) to fabricate SD and BC OSCs with different PT2 batches. As shown in Figure 5 and Table 2 (in the revised manuscript), the SD OSCs yielded high PCEs near 13%. However, the BC OSCs shows inferior PCE when the molecular weight of PT2 was increased. To further verify the results, we randomly synthesized two batches of PT2 by controlling the polymerization time. As shown in Supplementary Fig. 16 and Table 9 in the revised manuscript, when blended with Y6, the SD OSCs still yield $PCE \geq 16\%$, which is much higher than that of BC OSCs. These results strongly support the reliability and reproducibility of SD processing method in mitigating the polymer batch-to-batch problems.

(3) Expanding on that, AFM, GIWAXS, and RSoXS can be indirect evidence for the formation of more defined morphologies, but they only prove that SD approach results in better morphology for PT2 and Y6 blends and does not support their claims on general processing innovations. More systematic studies, such as the analyses of morphological and photovoltaic characteristics with CF (or CF/DIO) solvent washing of fibril layer comprising polymers with various PDIs and MWs, would be necessary.

Answer: Thanks for the comments. We investigated the morphological and photovoltaic characteristics of PT2 films with CF (or CF/DIO) solvent washing of fibril layer in the section of “Sequential-deposition and polymer batch tolerance” in the revised manuscript.

(4) It is questionable that the homogeneous vertical phase distribution (unlike inhomogeneous ones in other SD method works) is derived only from the fibril-based SD processing. The authors did not explain enough about Fig 2c in their difference with

Fig 2a and b, which can show that whether these characteristics are from molecular properties or processing methods.

Answer: Thanks for reviewer's suggestion. It is true that SD processing usually leads to serious vertical segregations, which, if properly used, can help to increase device performance. Such feature comes from the penetration of the second layer into the first layer. In the current case, we use optical method as an easy probe to compare the optical properties of BC and SD thin films, which yielded quite similar BHJ results (the slight difference will be discussed in the next paragraph). Thus we claim the SD and BC thin films have similar vertical material distribution, which is homogeneous. The optical absorption is a material property, and the morphology is correlated with processing method. The same recipe using different processing methods but with similar properties generally can be used to infer that the different processing methods do not lead to different feature that correlated with the probing property. Since the reviewer concerned about the segregation issue, we fabricated transfer printed layer-by-layer film to cross compare. The results were shown in Supplementary Fig. 7, there is a distinct vertical phase distribution. The results prove again that our SD film is a BHJ structure

Moreover, we alternatively use film-depth-dependent light absorption to investigate the crystalline ordering of PT2 and Y6 at different film-depth position. As shown in Figure R3, the absorption peak of PT2 is independent on film-depth, implying that its HOMO level (also hole transport level) is invariable along charge transport direction (film-depth direction). This negligible variation of transport level is beneficial to avoid trapping regions with low-energy localized states along transport direction, which is corresponding to the fact that PT2 is a high-performance donor material and its fibril network represents an ideal framework for bulk heterojunction. On the other hand, as compared with the SD (as-cast) films, the absorption peaks of Y6 of SD (optimum) and BC (optimum) films show a red-shift by 20 nm, demonstrating a higher crystalline ordering of Y6 molecules in the blends. This crystalline improvement occurs at every film-depth, contributing to the increased electron mobility.

Fig. R3 Film-depth-dependent light absorption spectroscopy of SD and BC films spin coated on the PEDOT:PSS/ITO substrate with different operating condition. **a** SD (as-cast) film. **b** SD (optimum) film. **c** BC (optimum) film. The spectra were vertically shifted and rescaled for clarity. The dashed vertical lines show the absorption peaks of PT2 and Y6. The sub-layer thickness for each spectrum is approximately 10 nm.

(6) In order to claim the innovative aspects of the work, it might be beneficial to show how a gradual scaling-up of processing deviates the morphological development in both SD and BC processing.

Answer: Thanks for the suggestion. We agree that a stepwise scaling-up of device fabrication can be useful concerning the large scaled processing in commercial purpose. Yet regarding to the scientific content, comparing the fabrication method of BC and SD addresses the fundamental issues we get interested. Spin-casting and layer-by-layer processing are fundamentally different regarding to morphology development mechanism. Increasing the thin film size might lead to morphology change. Such feature, if happened, should be correlated with very detailed environment change. We in this research simply use the lab setting to compare the structure and property correlations using SD and BC method. We did not think of scaling up in size in the current state.

Reviewer #2

The solution processing sequential deposition (SD) method has been paid attention owing to its advantages such as large-area printing and ease of film morphology control with high reproducibility. As aforementioned merits of SD, herein, they demonstrated the facilitation of fibril network strategy in the donor polymer layer within the SD films. For the depth analysis in the vertical direction as one of the crucial investigations of SD film, in addition, the film-depth-dependent light absorption spectroscopy clearly revealed the donor and acceptor composition of SD films, as well as exciton generation and dissociation within the photo-active layer. Moreover, the 0.8 cm² of large-area fabrication via slot-die printing with high PCE up to 14.6% ensures the worth of the SD system they found. Although the characterizations of their synthesized polymer PT2 are somewhat insufficient, they did a well-organized study about the SD system and the reviewer would recommend the publication of this paper after following minor revisions.

Answer: Thanks for the reviewer's positive comments for our work.

(1) The used PT2 polymer is not only random terpolymer of PBT1-C (Adv. Mater., 2018, 30, 1803045) and PBT1-C-2Cl (J. Mater. Chem. A, 2019, 7, 8055), but also firstly reported in this article. In the case of brand-new material, the full characterization data be always required. However, there is no characterization synthesized PT2 polymer such as NMR spectra, elemental analysis and so on. Especially, the molecular frontier orbitals of PT2 is not provided, but it might be changed compared to PBT1-C and PBT1-C-2Cl. They should properly characterized the new polymers like the papers (Nat. Commun. 2018, 1867). Besides, the authors need to provide some texts related on the random terpolymers (see the random review paper DOI:10.1002/adma.201804762).

Answer: Thanks for the suggestions. The characterization data of PT2 and the molecular frontier orbitals of PT2 have been provided in the supplementary information. We have also cited the suggested reference by the reviewer in the revised manuscript.

(2) They showed successful synthesis of different molecular weight of PT2 polymers and investigated the effects of molecular weight on the SD system. However, they didn't describe how to they control the molecular weight clearly. I recommended the author to supplementary describe the synthesis of polymer including aforementioned point 1.

Answer: Thanks for the suggestion. The detailed synthetic procedures have been provided in the Supplementary Information. Actually, the polymer molecular weight was controlled by simply changing the polymerization time.

Reviewer #3:

The authors describe the use of a conjugated polymer (PT2) and a small molecule acceptor (Y6) to reach very high power conversion efficiencies (16.5%) via a sequential deposition (SD) in which PT2 is deposited, followed by Y6. The main claim of the authors is that the SD method is advantageous compared to blend coating (BC) in which the PT2:Y6 mixture is deposited in one step.

The authors have submitted another paper (ref. 40) which describes, amongst others, PT2:Y6 blends for solar cells reaching a maximum efficiency of (15.6%). The efficiencies are similar and hence the main question to be answered is if the SD method is really advantageous compared to BC. There are already several reports on sequential deposition of semiconductor blends for organic solar cells, some are cited in the manuscript. The main advantage of a two-step procedure would be a significantly better performance or the fact that in SD less hazardous solvents could be used. In this example that is not the case. PT2 is deposited from chlorobenzene, followed by Y6 from chloroform (CF) with diiodooctane (DIO) as co-solvent. Then the layer is thermally annealed at 100 °C. In BC the blend is deposited from CF/DIO and also thermally annealed.

The authors argue in the abstract that the two-step SD eliminates the kinetic issue in morphology formation. But that is not what the data show. In fact in the manuscript the

authors assume a certain amount of PT2 fibrils is dissolved during the second layer casting and demonstrates that presence of DIO is needed to recover the PT2 fibril structure. It is difficult to see the advantage of a two-step procedure if the second step is identical to the procedure that would be used in a one-step (BC) deposition. There are indeed small differences between SD and BC deposited films, but nothing dramatic.

Answer: Thanks for the comments. We got excited with the SD method since the batch-to-batch variation issue can be eliminated. However, we might not stress this point enough that leads to misunderstanding. As the reviewer mentioned, a PCE of 15.6% can be achieved for BC OSCs based on PT2 ($M_n = 45$ k) in the submitted paper. However, the PCEs of BC devices significantly decreased with the increase of the PT2 molecular weight (see Table 2 in the revised manuscript), which shows the limitation of BC in OSC preparation. In contrast, the SD OSCs showed performance insensitivity to PT2 batch-to-batch variations, with much better statistics comparing to BC OSCs. We think this finding is really important that could release the enormous chemistry efforts in finding the right material of the right batch. We might not stress this key enough in initial submission. In the revised version, we extend SD processing to several other systems, such as IT-4F to show generality. As shown in Figure 5 and Table 2 (in the revised manuscript), the SD OSCs yielded high PCEs near 13% regardless of PT2 choice. However, the BC OSCs shows inferior PCE when the molecular weight of PT2 was increased. To further verify the results, we randomly synthesized two batches of PT2 by controlling the polymerization time. As shown in Supplementary Fig. 16 and Table 9 (in the revised manuscript), when blended with Y6, the SD OSCs keep high PCEs $\geq 16\%$, which is much higher than that of BC OSCs. The results indicated fundamental differences between SD and BC. We assiduously sought the scientific reasons by employing multiple characterizations, and constructed a plausible structure-property relationship that help to assure the advantage of SD strategy theoretically. The manuscript is revised in a large extent, which should be more accurate and obvious in finding the core information much easier.

(1) Line 139: 3.152 cm² should be 3.152 mm²

Answer: Thanks for the suggestion. This mistake was corrected in the revised manuscript.

(2) Line 142: they reach 15.5% in ref 40 for the BC device, now 15.0%, why is it less?

Answer: Thanks for the comment. As shown in Fig. 5 and Table 2 in the revised manuscript, the BC OSCs show performance sensitive to PT2 batch variation. PCEs of ~15.5% could be achieved for BC OSCs fabricated with PT2 with molecular weight around 45K. The PCE was decreased to 15% when PT2 with higher molecular weight of 57K.

(3) Line 162: The authors present a good stability for the SD OSC, but there is no comparison with BC OSC devices. Why would it be an advantage?

Answer: Thanks for the comment. We compared with the ambient stability of the optimum SD and BC OSCs. As shown in Supplementary Fig. 3, after exposed in air for more than 300 hours, the SD device maintained 98.8% of its initial PCE, which is higher than that of the BC device. The results indicate that the prescribed fibril network BHJ morphology in SD fabrication is superior than that of BC processing, which results from the simple thermodynamics that materials interaction is less entangled with crystallization that diffusion of Y6 into amorphous PT2 creates better quality n-type regions.

(4) The discussion on the hole and electron mobilities show that all are of the same magnitude. The SCLC method is not very accurate and the three significant digits given are not realistic. If they want to maintain their numbers, the authors should study the SCLC behavior at different thickness, prove the L-3 behavior and show that the mobility is constants. Now they fit a very small part of J-V curve for an unknown thickness where by chance the current density scales with square of the voltage. I know it is done in many papers, but it is not convincing. Supp. Fig 5, lacks a vertical scale.

The current density of the BC electron-only device is higher than that of the BC device but the mobility is less.

Answer: Thanks for the comments. It is a good idea to study the SCLC behavior at different thickness for SD OSCs. However, in fact, it is difficult to guarantee a certain weight ratio of polymer donor and acceptor when the active layer thickness increased for SD OSCs. Instead, we performed the photo-CELIV measurements, which is also widely used to measure the charge carrier mobility in OSCs. As show in Supplementary Fig. 5, Fig. 17 and Table 10, the carrier mobilities of the SD devices were almost twice of the BC devices, agreeing well with the SCLC results. In addition, we retested the device mobility by SCLC method, and averaged the charge mobility from 10 dependence devices as shown in Supplementary Fig 4 and Table 4.

(5) Concerning the ideality factor the optimum SD device showed a slope of 1.25 kT/q , while the optimum BC device has slope of 1.41 kT/q . The authors conclude that the SD method suppresses the trap-assisted recombination in the blends compared to the BC method. This is a too simple line of a reasoning. First the V_{oc} of the two devices is the same, hence in terms of non-radiative recombination, the SD is not better than the BC. The authors forget that an ideality factor 1 is not only expected form bimolecular recombination, but also for surface recombination. The ideality factor itself is not a very meaningful parameter in practice.

Answer: Thanks for the comments. We agreed with the reviewer's suggestion. The surface recombination would also affect the ideality factor. But it is not a significant loss channel in organic solar cells, especially for those with relatively high fill factors and power conversion efficiencies (Adv. Mater. 2019, 31, 1903868). To avoid misunderstanding the charge recombination behavior, we deleted the results of V_{oc} at different light intensities in the revised manuscript.

Reviewers' Comments:

Reviewer #1:

Remarks to the Author:

The authors tried to address the reviewers' concerns in the revised manuscript by additional experiments for more robust proof of the concept as well as for the acquisition of generality. There are still several minor technical questions in regard to the essence of their claim on the robust and reliable processing. Once these issues are satisfyingly addressed, the reviewer can agree with the publication of this paper in Nature Communications.

Technical issues are as follows

1. The authors showed in the response to reviewers' comments that the CF/DIO solvent washing has minimal impact on the photovoltaic performance. It is a very promising result toward the reliability of device fabrication. However, it also further raises concerns about the role of the amorphous regions. It is true that the authors tried to move away from the MW and PDI issues by going over certain MW in polymer, but still, non-unity PDI means there are low MW region that interacts more freely with the secondary-deposited acceptor layers to a different degree depending on the choice of acceptor, as they also claim in the manuscript. Of course, it is reasonable that the amorphous region can become beneficial in achieving higher performance through constructing multiscale morphology, but it also means that the relative amount of amorphous/crystalline and how the amorphous region interacts with acceptors affect the performance of SD fabricated OPVs. These issues lead to the concerns about their strongest claim on batch-to-batch invariance. It would be the best to test in different PDIs or narrower PDIs (although I understand it is very difficult without specialized equipment), it would be sufficient to show that you can minimize the parasitic impact of the amorphous regions, however losing some performance, and rely solely on the crystalline network for the purpose of achieving the reliability.

Reviewer #2:

Remarks to the Author:

The authors addressed properly all the reviewers' comments, I believe that the current revised manuscript reached to high level quality of Nature communications, thus I recommend it to be published in the Nature communications.

Reviewer #3:

None

Reply to reviewer's comments (adma. 201905645):

Reviewer # 1

The authors tried to address the reviewers' concerns in the revised manuscript by additional experiments for more robust proof of the concept as well as for the acquisition of generality. There are still several minor technical questions in regard to the essence of their claim on the robust and reliable processing. Once these issues are satisfyingly addressed, the reviewer can agree with the publication of this paper in Nature Communications.

The authors showed in the response to reviewers' comments that the CF/DIO solvent washing has minimal impact on the photovoltaic performance. It is a very promising result toward the reliability of device fabrication. However, it also further raises concerns about the role of the amorphous regions. It is true that the authors tried to move away from the MW and PDI issues by going over certain MW in polymer, but still, non-unity PDI means there are low MW region that interacts more freely with the secondary-deposited acceptor layers to a different degree depending on the choice of acceptor, as they also claim in the manuscript. Of course, it is reasonable that the amorphous region can become beneficial in achieving higher performance through constructing multiscale morphology, but it also means that the relative amount of amorphous/crystalline and how the amorphous region interacts with acceptors affect the performance of SD fabricated OPVs. These issues lead to the concerns about their strongest claim on batch-to-batch invariance. It would be the best to test in different PDIs or narrower PDIs (although I understand it is very difficult without specialized equipment), it would be sufficient to show that you can minimize the parasitic impact of the amorphous regions, however losing some performance, and rely solely on the crystalline network for the purpose of achieving the reliability.

Answer: Thanks for the comments. We agree with the reviewer that the amorphous region is an important part in bulk heterojunction (BHJ) morphology, which, however, is quite difficult to characterize. We can briefly comment on the PDI issue as the reviewer concerned. It is known that conjugated polymers used in organic BHJ solar cells suffer from serious batch-to-batch variation issues. Such problem comes from both materials side and processing side, which in combination affect the BHJ morphology. For

conjugated polymers that reach certain molecular weight, we noticed that the materials crystallization becomes quite robust in the neat films, which does not change largely upon molecular weight change. We developed the SD device fabrication method, trying to avoid the complicated materials interaction and to rely on material crystallization, by which we can avoid the polymer batch-to-batch variation issues. Success can be seen from the results presented in Table 1. The amorphous polymer and acceptor interaction in SD processing is much weaker than that in BC solution processing, since the nonequilibrium film formation process can be avoided. The morphology formation is mainly dictated by the difference of amorphous polymer and crystalline polymer dissolution upon processing solvent. Direct supporting results can be seen from GIWAXS results of improved crystallization in SD processed films. Improved crystalline content leads to reduced amorphous part, and thus detailed materials interaction is not as important as in blend casting method. In device characteristics, better polymer crystallinity was seen from improved carrier mobilities, indicating the crystalline fiber framework in SD processing methods can be better than conventional blend casting, which also supports the argument that SD processing shifts the previous focus of complicated materials interaction to polymer crystallization. From the processing side, this is much easier to control since the second layer casting time can be easily manipulated.

The exact amorphous content is hard to quantify, and we could only rely on final thin film crystallinity and device performance to consolidate a conclusion. As shown in Table 1, we tested 5 batches of PT2 with different molecular weights and PDIs, including 2 batches synthesized randomly at different polymerization time. We can find that the average PCEs are ranging from 15.9% to 16.4% for SD OSCs, much better than PCE variations (from 13.2% to 15.3%) in BC devices, indicating that the amorphous regions have minor influence on the photovoltaic performance in SD OSCs, which is also highlighted in the revised manuscript. We totally agree with the reviewer that detailed MW and narrower PDI polymers could help to fully resolve the details, but that is far from our capability.

Table 1 Summary of device parameters of SD and BC OSCs with different PT2 batches						
Blend	M _n and PDI	Operating Conditions	V _{oc} (V)	J _{sc} (mA/cm ²)	FF (%)	PCE (%) (average) ^a
PT2:Y6	45k (2.33)	SD	0.82	26.3 (25.8) ^b	74.7	16.1 (16.0±0.1)
		BC	0.82	25.7 (25.3)	74.1	15.6 (15.3±0.3)
	57k (2.05)	SD	0.83	26.7 (26.3)	74.4	16.5 (16.3±0.2)
		BC	0.83	26.3 (25.6)	68.9	15.0 (14.7±0.2)
	91k (2.06)	SD	0.82	27.1 (26.4)	71.8	16.0 (15.9±0.2)
		BC	0.81	26.5 (26.0)	62.4	13.4 (13.2±0.1)
	random batch I	SD	0.83	27.0 (26.3)	73.2	16.3 (16.3±0.0)
		BC	0.82	26.0 (25.5)	71.4	15.3 (15.1±0.2)
	random batch II	SD	0.83	27.4 (26.7)	73.7	16.7 (16.4±0.3)
		BC	0.82	26.3 (25.7)	67.2	14.5 (14.4±0.1)

^a The average parameters were calculated from 10 independent cells.
^b EQE values.

Reviewer #2:

The authors addressed properly all the reviewers' comments, I believe that the current revised manuscript reached to high level quality of Nature communications, thus I recommend it to be published in the Nature communications.

Answer: We appreciate the reviewer for the very positive recommendation of our work.

Reviewers' Comments:

Reviewer #1:

Remarks to the Author:

The authors addressed the reviewers' concerns on the role of the amorphous region and the batch-to-batch variations. The technical advancement that could be achieved by this work could be of high importance in the field of organic photovoltaics, especially since the field is evolving so quickly toward the actual up-scale productions. The reviewer would gladly recommend the publication of this article in Nature Communications.

Reviewer #1 (Remarks to the Author):

The authors addressed the reviewers' concerns on the role of the amorphous region and the batch-to-batch variations. The technical advancement that could be achieved by this work could be of high importance in the field of organic photovoltaics, especially since the field is evolving so quickly toward the actual up-scale productions. The reviewer would gladly recommend the publication of this article in Nature Communications.

Answer: We appreciate the reviewer for the very positive recommendation of our work.